# Effects on the Physical Functioning of Two Exercise Interventions in Patients with Multiple Myeloma: A Pilot Feasibility Study

**DOI:** 10.3390/cancers16091774

**Published:** 2024-05-04

**Authors:** Jens Hillengass, Michaela Hillengass, Janine M. Joseph, Kristopher Attwood, Rikki Cannioto, Hillary Jacobson, Carolyn Miller, Bryan Wittmeyer, Kirsten Moysich

**Affiliations:** 1Department of Medicine—Myeloma, Roswell Park Comprehensive Cancer Center, Elm and Carlton Streets, Buffalo, NY 14263, USA; 2Department of Cancer Prevention and Control, Roswell Park Comprehensive Cancer Center, Elm and Carlton Streets, Buffalo, NY 14263, USA; 3Department of Biostatistics and Bioinformatics, Roswell Park Comprehensive Cancer Center, Elm and Carlton Streets, Buffalo, NY 14263, USA; 4Department of Physical Therapy, Roswell Park Comprehensive Cancer Center, Elm and Carlton Streets, Buffalo, NY 14263, USA

**Keywords:** physical activity, physical function, pain, quality of life, multiple myeloma

## Abstract

**Simple Summary:**

Individuals living with multiple myeloma are likely to have bone destruction as a consequence of their disease, leading healthcare providers to be reluctant to recommend physical activity. The aim of this prospective trial was to assess the feasibility of six-month strength training and walking interventions in patients with multiple myeloma. Various assessments of physical function and pain were performed at multiple timepoints throughout the trial. Participants saw improvements in mobility, leg strength, aerobic capacity, and endurance, with more pronounced and sustained improvements in the strength training arm, particularly in leg strength. This small trial showed the feasibility and benefits of both strength training and walking interventions in patients living with multiple myeloma. A larger trial extending these findings is underway at our institution.

**Abstract:**

Because of the high prevalence of bone destruction in patients with multiple myeloma (MM), physical exercise is oftentimes discouraged by healthcare providers. The goal of this prospective trial was to investigate the feasibility of two six-month exercise interventions in patients with MM (*N* = 42): a remotely prompted home-based walking intervention or a supervised strength training intervention. Physical function and pain were assessed with the Activity Measure for Post-Acute Care (AM-PAC) Basic Mobility Short Form raw score, a six-minute walk test (6 MWT), a 30-second sit-to-stand test (30 SST), a timed up-and-go (TUG) test, a visual analog scale (VAS) for pain, handheld dynamometer tests, heart rate at rest, blood oxygen saturation at rest, and body mass index. No intervention-related serious adverse events were observed. Adverse events mostly affected the musculoskeletal system. In the resistance training group (*n* = 24), patients showed significant improvements in AM-PAC, TUG, 6 MWT, and 30 SST, with all effects but the 6 MWT sustained six months after the intervention. The walking group (*n* = 18) saw improvements in the AM-PAC, TUG, 6 MWT, and 30 SST, with a sustained change in the AM-PAC and TUG. This trial shows the feasibility of both exercise interventions with a sustained beneficial effect on the physical functioning of a six-month strength training intervention and, to a lesser extent, a six-month unsupervised walking intervention. A larger study building on these findings is currently underway.

## 1. Introduction

The main symptoms of multiple myeloma (MM), a hematologic disease caused by the proliferation of malignant plasma cells in the bone marrow, are cytopenia, renal impairment, and bone destruction caused by an over-activation of osteoclasts [1]. The latter leads to musculoskeletal pain, instability, fractures, and reduced mobility, as well as an oftentimes immense influence on the quality of life [2]. Numerous treatment options have been developed and are in clinical use. Cornerstones of MM therapy are combination treatments with proteasome inhibitors (PIs), immunomodulatory drugs (IMiDs), selective inhibitors of nuclear export (SINE), and monoclonal antibodies (MoAbs), oftentimes combined with high-dose chemotherapy and autologous stem cell transplantation (ASCT) [3,4]. More recently, chimeric antigen receptor T-cells (CAR-Ts) and bispecific T-cell engagers (BiTEs) have been approved [5,6]. The above-mentioned therapies have increased the overall survival of MM patients significantly [7]. Therefore, in addition to the quantity of life, quality of life has become an increasingly important aim of MM patients, providers, and researchers. Physical therapy (PT) and physical activity, in general, can contribute to the overall well-being of patients, especially if their skeletal systems are affected by cancer [8,9]. However, out of fear of further bone damage, MM patients are oftentimes advised not to participate in any such activities, even though many of them report a strong desire to do so [10]. Certain MM therapies, especially ASCT [11] and CAR-T [12], are associated with some level of toxicity, and patient frailty is considered a risk factor for complications, independent of the effect of age [13], sometimes leading to the exclusion of patients from these therapies or lower doses/shorter durations of treatment and worse outcomes [14,15,16,17]. Although a common characteristic of MM patients, particularly in the relapsed/refractory setting [16], frailty is a somewhat influenceable factor and can be improved through interventions like PT and physical exercise [18], potentially paving the way to better treatment outcomes. In the setting of ASCT, such interventions have been shown to shorten the length of stay and reduce the number of readmissions [19].

Here, we report the results of a pilot study in patients with MM utilizing supervised strength training or an unsupervised behavioral walking intervention to investigate feasibility and adherence as the primary endpoints. The secondary endpoints included functional status as evaluated via a PT assessment, immunological markers, and patient-reported quality of life.

## 2. Materials and Methods

### 2.1. Patients and Setting

From January 2020 to August 2021, a total of 42 patients were enrolled in and began this prospective, non-randomized interventional pilot study, drawn from the population of patients with multiple myeloma receiving outpatient care in the Myeloma Clinic of Roswell Park Comprehensive Cancer Center (Roswell Park) in Buffalo, New York. The study was approved by Roswell Park’s Institutional Review Board (IRB). The patients provided informed consent for all the study-related procedures and publication of the anonymized results. All procedures performed in this study were in accordance with the ethical standards of Roswell Park’s IRB and with the Belmont report. Since the trial included an in-person component and the COVID-19 pandemic occurred during this period, the study was terminated early, and a new trial, which is currently enrolling patients, was initiated, offering virtual guided workouts only (NCT05312255).

The patients were pre-screened by their treating physician to eliminate those with functional limitations or other health problems that would have prevented them from safely participating. Patients who passed pre-screening were approached and screened for participation. Before beginning the intervention, patients who were deemed eligible had their most recent imaging results reviewed in an interdisciplinary tumor board with at least an oncologist, a radiologist, and a neurosurgeon present to assess the bone and especially spinal stability. The tumor board further required some patients to have a neurosurgery or orthopedic stability consult prior to clearance. The patient was either cleared for participation, cleared with limitations, or deemed ineligible to participate. Most patients were cleared by the tumor board with no limitations (59.5%) or cleared with no limitations after a neurosurgery or orthopedic consult (16.7%). Fewer than one quarter were cleared with limitations on the weight or movements allowed (23.8%), and no patients were deemed ineligible to participate. The number of patients and available assessments at each timepoint are shown in Figure 1 and are described in the Section 3.

### 2.2. Intervention

Patients assigned themselves into one of two active interventions: (1) a six-month in-person strength training program in our PT department with an American College of Sports Medicine (ACSM)-certified personal trainer in small group sessions, training 60 min twice a week and targeting all major muscle groups (strength training group, STG); and (2) a six-month at-home behavioral intervention using wearable fitness trackers (Fitbit Inspire HR, Fitbit LLC, San Francisco, CA, USA) with regular remotely delivered prompts motivating patients to achieve 150–300 active minutes per week (walking group, WG). The design, which was non-randomized, allowed participants who wanted to be a part of the study but did not, for whatever reason, want to be in the in-person STG (e.g., distance from the hospital or COVID-19 concerns) to select the remote WG.

### 2.3. Safety

As part of the trial monitoring, the participants were closely watched for the occurrence of adverse events (AEs) and serious adverse events (SAEs) that may or may not have been related to the intervention. Descriptions of the events, grades, and relationships to the interventions were tracked and reported as appropriate to the Roswell Park IRB. The events were categorized based on the Medical Dictionary for Regulatory Activities (MedDRA) primary system organ class (SOC)—the highest level in the MedDRA hierarchy–from the Common Terminology Criteria for Adverse Events (CTCAEs) Version 5.0 [20]. Appendix A include the descriptions of the AE/SAE categories (S1) and grade/relationship structures (S2). For the purposes of this manuscript, we have provided the total number of AEs, grade 3 or higher AEs, and AEs that were at least possibly related to the intervention. We have also provided incident-level details on all SAEs, even though none were related to the intervention.

### 2.4. Physical Therapy Assessment

The PT assessment was planned at the baseline, after three months of the intervention, at the end of the intervention (six months), and three and six months after the end of the intervention. It included the following tests: the 18-item Boston University Activity Measure for Post-Acute Care (AM-PAC) Basic Mobility Outpatient Short Form raw score and the timed up-and-go (TUG) test (seconds [s]) to assess the mobility, a six-minute walk test (6 MWT, meters [m]) to measure the aerobic capacity and endurance, a 30-s sit-to-stand test (30 SST, repetitions [reps]) for the assessment of leg strength and endurance, a visual analog scale (VAS) for pain measurement, a handheld dynamometer (left and right [lbs.]) for the strength assessment, a heart rate at rest test (beats per minute), a blood oxygen saturation at rest test (%), and body mass index (kg/m^2^) [21,22,23,24,25,26,27].

### 2.5. Endpoints

The primary endpoint of feasibility was defined as the proportion of enrolled patients who were still in the study at the end of the six months. The secondary endpoint of adherence was defined in the STG as the proportion of patients remaining in the study at six months who completed 80% or more of the strength training intervention activities. For the WG, it was defined as the proportion of patients whose average daily step counts each week were at or above their goal at least 20 out of 26 weeks or 80% of the weeks.

As exploratory endpoints, it was investigated if the patient-reported outcomes changed over the course of the study and if there was an improvement in physical functioning according to the aforementioned PT assessment or changes in the immune cell subsets (Joseph et al., in press) [28].

### 2.6. Statistical Methods

The key demographic and clinical characteristics of the intervention groups were compared using Student’s *t*-tests for differences in the means or Chi-squared tests for differences in proportions. We compared the changes in the physical function outcomes over time and between the groups using the generalized estimating equation (GEE) modification of linear regression to account for correlations within individuals over time. In all comparisons, a value of *p* < 0.05 was considered significant. The effect sizes between the timepoints were calculated as the mean divided by the standard deviation of the subject-level differences between the timepoints. All analyses were performed using SAS 9.4 for Windows (SAS Institute, Cary, NC, USA).

### 2.7. Data Availability

The original datasets are available upon reasonable request from the corresponding author.

## 3. Results

### 3.1. Descriptive Characteristics

The characteristics of the participants (*n* = 42) are shown in Table 1. The participants were a mean of 63.2 years of age at consent with a mean body mass index (BMI) of 30.3 kg/m^2^. The study sample was 61.9% female, and 85.7% self-identified with the non-Hispanic White race. Most participants had non-active disease (90.5%), had experienced a complete treatment response (61.9%), and had an Eastern Cooperative Oncology Group (ECOG) performance status of 0 (“fully active”) (61.9%). At baseline, the patients were on various combinations of treatment, including immunomodulatory drugs (IMiDs) without (45.2%) or with monoclonal antibodies (MoAbs) (16.7%), proteosome inhibitors (PIs) (11.9%), other regimens (11.9%), or no current treatment/surveillance (14.3%). There were no differences by treatment arm in these characteristics.

### 3.2. Feasibility and Adherence

Figure 1 is a flowchart showing the number of patients for each timepoint. Of a total of 87 patients who were eligible to participate, 43 (49%) signed the consent form and were enrolled. The reasons for the refusal of participation in 44 patients were distance to the treatment center or issues with transportation in 23 patients, no interest either in clinical trials or exercise in 11 patients, not enough time in seven, and other reasons in three patients. One patient who signed the consent form did not complete a baseline assessment and was excluded from the analysis.

The 42 enrolled patients who began the intervention were in the study for a median of 6 months (range: 1–15). Within the STG, the major feasibility challenge was that the COVID-19 pandemic caused delays and interruptions for five patients, three of whom withdrew from the study. There were also delays and interruptions caused by other medical issues, including disease progression in six patients, two of whom withdrew from the study. Non-compliance led to early termination in two patients, and two other patients withdrew because they changed their minds about participating. Within the WG, technical difficulties with syncing data between the participants’ Fitbits and our research database impacted the reporting of measures such as active minutes and steps.

The proportion of enrolled patients who were still in the study at the end of six months—our primary feasibility endpoint—was 77% overall, 79% in the STG, and 74% in the WG (*p* = 0.67).

Adherence to the requirements of the study was higher in the STG than in the WG. Within the STG, 100% of the patients who were still in the study at the six-month visit (79% of the enrolled patients) performed all the required strength training sessions. The aforementioned delays caused the STG intervention period to stretch out beyond the anticipated six-month period (7–15 months, median = 8 months); only 32% of the patients completed at least 80% of the training sessions within six months of beginning the intervention. Due to technical difficulties with syncing data between the participants’ Fitbits and our research database, we gathered more limited data on adherence within the WG. In the WG, 16 patients were available for compliance analysis, including those who did not complete the intervention. Interestingly, only one patient met their active minute goal for at least 80% of the weeks, while three patients met their active minute goals for 50% of the weeks. For all subjects combined, 28% of the active minute goals of the individual person-weeks were met (35% in the first half and 19% in the second half of the intervention). Adherence to the required physical function assessments was excellent: assessments were performed for 95% of the subject timepoints from the baseline to six months post-intervention (ST = 99%; WG = 90%). 

### 3.3. Safety

Table 2 provides summary details on the adverse events (AEs) reported, and Table 3 provides detailed information on the specific serious adverse events (SAEs). Most (78%) adverse events (AEs) and all serious adverse events (SAEs) were unrelated or unlikely to be related to the intervention. Of the AEs that were considered to be possibly, probably, or definitely related to the intervention, the majority (18 of 24) were musculoskeletal symptoms, of which none were grade 3 (severe) or higher. There were also four instances of dizziness related to the intervention, captured in the “nervous system disorders” category. No fractures or other skeletal-related events were reported.

### 3.4. Physical Functioning

Improvements were seen in multiple functional assessments over the course of the trial, as shown in Figure 2. The significant findings are as follows, with the mean ($\bar{X}$) and effect size (d) of the timepoint differences shown in parentheses. Among both the STG and the WG, the basic mobility score (AM-PAC raw score) improved significantly during the intervention (STG: $\bar{X}$ = 5.1, d = 1.14; WG: $\bar{X}$ = 4.0, d = 0.87) and remained improved throughout the follow-up period, particularly for the WG (STG: $\bar{X}$ = 3.9, d = 0.64; WG: $\bar{X}$ = 4.7, d = 1.25). Likewise, the timed up-and-go (TUG) test improved in patients in both the STG and the WG (STG: $\bar{X}$ = −1.4 s, d = −1.03; WG: $\bar{X}$ = −1.2 s, d = −1.29) and was sustained throughout the follow-up period (STG: $\bar{X}$ = −1.0 s, d = −0.76; WG: $\bar{X}$ = −0.8 s, d = −0.66). The STG and WG showed significant improvements during the intervention in the six-minute walk test (6 MWT) (STG: $\bar{X}$ = 39.3 m, d = 0.59; WG: $\bar{X}$ = 43.1 m, d = 0.71), but it was not sustained in either group during follow up. The STG improved in the 30-second sit-to-stand test (30 SST) ($\bar{X}$ = 4.4 rep, d = 1.24), which persisted through to the follow-up ($\bar{X}$ = 4.1 rep, d = 1.33), while the walking group saw a smaller improvement in this measure during the intervention ($\bar{X}$ = 1.8 rep, d = 0.80), which was not sustained. Improvements in pain using the visual analog scale (VAS) were non-significant for both groups during the intervention period, although the change from the baseline to six months post-intervention was significant for the STG ($\bar{X}$ = −9.4, d = −0.51). No significant changes were seen for the handheld dynamometer (left and right), resting heart rate, blood oxygen saturation at rest, or body mass index.

## 4. Discussion

This pilot study of exercise in patients with MM examined matters of feasibility, adherence, and changes in patient-reported outcomes, immune profiles (Joseph et al., in press) [28], and physical functioning. The study was shown to be feasible in our patient population, and there was excellent adherence to the six-month intervention requirements in the supervised strength training group and somewhat less adherence in the unsupervised walking group. Several parameters of physical functioning were improved after the six-month exercise interventions. Importantly, leg strength (30 SST), endurance (30 SST, 6 MWT), functional mobility (AM-PAC, TUG), and pain (VAS) improved significantly, especially in the STG. This is clinically relevant since physical function and frailty are factors influencing not only the choice of treatment but also the clinical outcomes. In fact, the International Myeloma Working Group (IMWG) has suggested a frailty score based on age, comorbidities, and activities of daily living (ADL) to assess MM patients at first diagnosis, and it has been shown that less fit patients had significantly more non-hematologic adverse events, discontinued treatment more often, and had a lower progression-free and overall survival [17]. In the current study, we measured the changes in the AM-PAC score because it includes functional mobility in the context of ADLs. Future planned studies will have the infrastructure to allow for broader frailty measurements, such as those proposed by the IMWG, which will allow for the evaluation of the impact of exercise regimens on objectively measured frailty.

The two intervention groups showed improvements in all measures with a similar magnitude of effect, with the exception of the 30 SST, which had markedly more pronounced and sustained improvement in the STG. However, the small sample sizes in both groups of this pilot study, as well as the effects of the COVID-19 pandemic on participation, limited the interpretability of the results comparing the two interventions, and since this was a feasibility study, we did not offer a control group for comparison. Furthermore, the more hands-off intervention in the WG led to lower adherence in that group. The lower observed adherence in the unsupervised group was not unexpected. Although a meta-analysis of exercise interventions in older adults showed that supervised interventions had non-significantly higher adherence than unsupervised interventions, the authors suggest that the adherence in unsupervised training could have been over-estimated due to being measured through self-reported diaries and by the fact that about half of the unsupervised studies actually included some supervision [29]. In our study, adherence in the unsupervised arm should not have been subject to those biases, as it was measured objectively, i.e., not via self-reporting, but technical issues with syncing the devices limited the conclusions that could be drawn from these data.

The exercise interventions in this study were designed with a high safety level in mind. The participants in both groups had an initial assessment by a tumor board with an oncologist, a radiologist, and a neurosurgeon in attendance, followed by an assessment by a physical therapist. In addition, the patients in the STG performed all exercises in person in the PT department of our institution under the supervision of a specialized exercise trainer. A recent review of the feasibility and safety of exercise interventions in patients with MM showed that there were very few intervention-related AEs and no SAEs in seven trials including 563 participants [30]. In our trial, we also saw no SAEs or grade 3 or higher AEs related to the intervention.

Our study adds to the collective understanding of the effects of exercise in patients with MM. To date, there are limited data on this subject, and most published studies have shown only modest effects [31]. In the peri-ASCT setting in MM patients, an intervention that combined stretching, walking, and a limited resistance training routine seemed to improve the effects of erythropoietin, reducing the need for blood transfusions, but did not show any further effects on physical functioning [32]. In two more recent randomized trials, including 58 MM patients post-ASCT and 100 patients with newly diagnosed disease, respectively, no effects were seen in the exercise groups, both of which contained aerobic and resistance training, compared to the control groups [33,34]. In the largest study to date, which enrolled 187 patients with newly diagnosed MM in the intervention arm (an individualized home-based aerobic and strength resistance training program), there were improvements in sleep, fatigue, and general physical aerobic capacity assessed using the 6 MWT [35].

The fact that our study showed several intraindividual improvements compared to most of the other trials might be explained by several factors. Our trial was the longest intervention to date, with patients being active for six months or 26 weeks (comparable to another positive study by Coleman et al. [36]) as compared to 12–18 weeks in most other trials. Furthermore, all our patients were in a steady disease state at the time of enrollment. The STG underwent a supervised intervention, which was similar to the largest study (again by Coleman), which also showed positive effects on functional markers like the 6 MWT, supporting the hypothesis that a certain duration of intervention is necessary to achieve effects. Another encouraging finding in our data was that the patients seemed to maintain their improvements after three and six months post-intervention. Anecdotally, the STG patients reported that they were sad that the intervention ended and that they were very interested in continuing. They also stated that the regularly scheduled exercise sessions helped them to remain compliant for a prolonged period of time. 

A major goal of lifestyle interventions is a sustained effect on quality of life, functional performance, and even clinical outcomes. Our study is the first to show changes in physical function that persisted after the end of the intervention. It is unclear, however, if this persistent benefit was related to the length/type of intervention or to the characteristics of our study sample, which had mostly stable disease in deep remission.

The strengths of the current study included the long duration of the intervention and post-intervention follow-up period, especially relative to other exercise interventions in cancer patients; the supervised nature of the strength training intervention, which ensured that patients were performing the exercises safely and properly; and the collection of both objectively measured and patient-reported outcomes. The primary limitations included the small sample size, which reduced the power to detect significant changes in physical functioning and other measures, as well as limiting comparisons between the two intervention arms; a follow-up period which was too short to enable an analysis of the effects on progression or survival in our patients; and the lack of a control arm. In spite of these limitations, improvements in physical function were observed in both the strength training and walking arms and were largely sustained after the intervention was over. The ongoing study at our institution (NCT05312255), which includes a virtual supervised strength training regimen very similar to that used in this study and with some of the same endpoints described herein, is on track to achieve a larger accrual of patients, which will clarify some of the findings of the current study. Additionally, a waitlist control group is being considered for future trials being planned.

In conclusion, this trial has confirmed the feasibility and safety of two exercise interventions and their efficacy on markers of physical functioning. It has paved the way for a larger prospective trial, currently underway at our institution, which includes additional endpoints such as bone strength and immune cell subsets in the peripheral blood. We also gained important insights into unsupervised behavioral interventions, as the monitoring—or lack thereof—of the WG was an obvious point of weakness in that arm. More sophisticated methods of measuring activity might provide more reliable results but will also be more expensive.

## 5. Conclusions

In conclusion, this trial has confirmed the feasibility and safety of two exercise interventions and their efficacy on markers of physical functioning. It has paved the way for a larger prospective trial, currently underway at our institution, which includes additional endpoints such as bone strength and immune cell subsets in the peripheral blood. We also gained important insights into unsupervised behavioral interventions, as the lack of monitoring of the WG was an obvious point of weakness in that arm. More sophisticated methods of measuring activity might provide more reliable results but will also be more expensive.

## Figures and Tables

**Figure 1 cancers-16-01774-f001:**
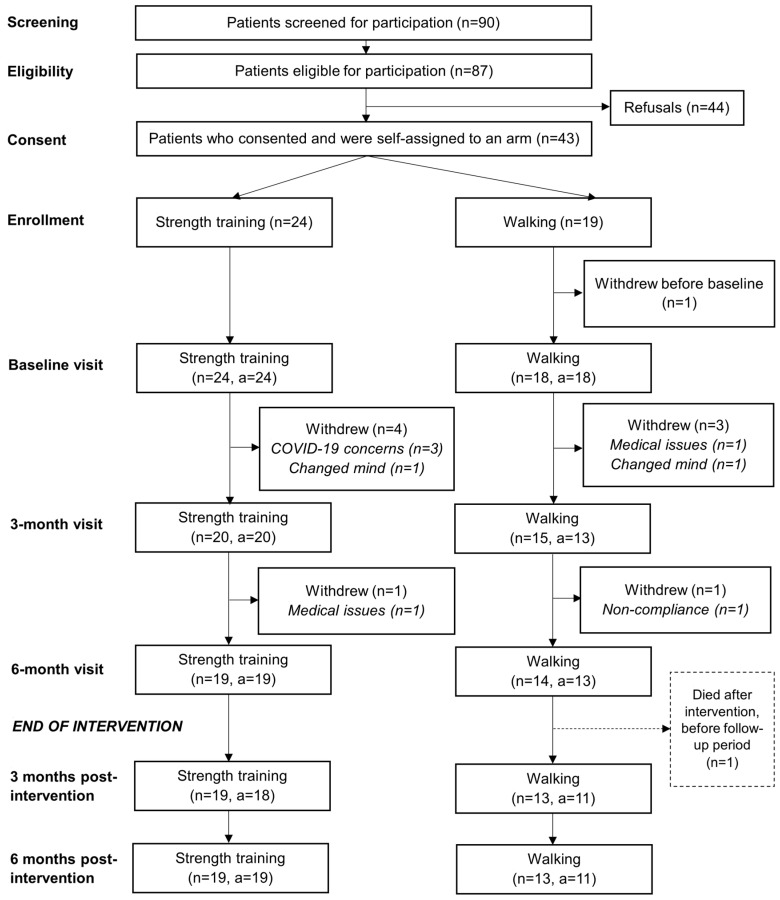
Flowchart of the numbers of patients at each timepoint and by arm from screening through to six months post-intervention. Abbreviations: *n* = number of patients at each timepoint; a = number of physical function assessments available from the participating patients at each timepoint.

**Figure 2 cancers-16-01774-f002:**
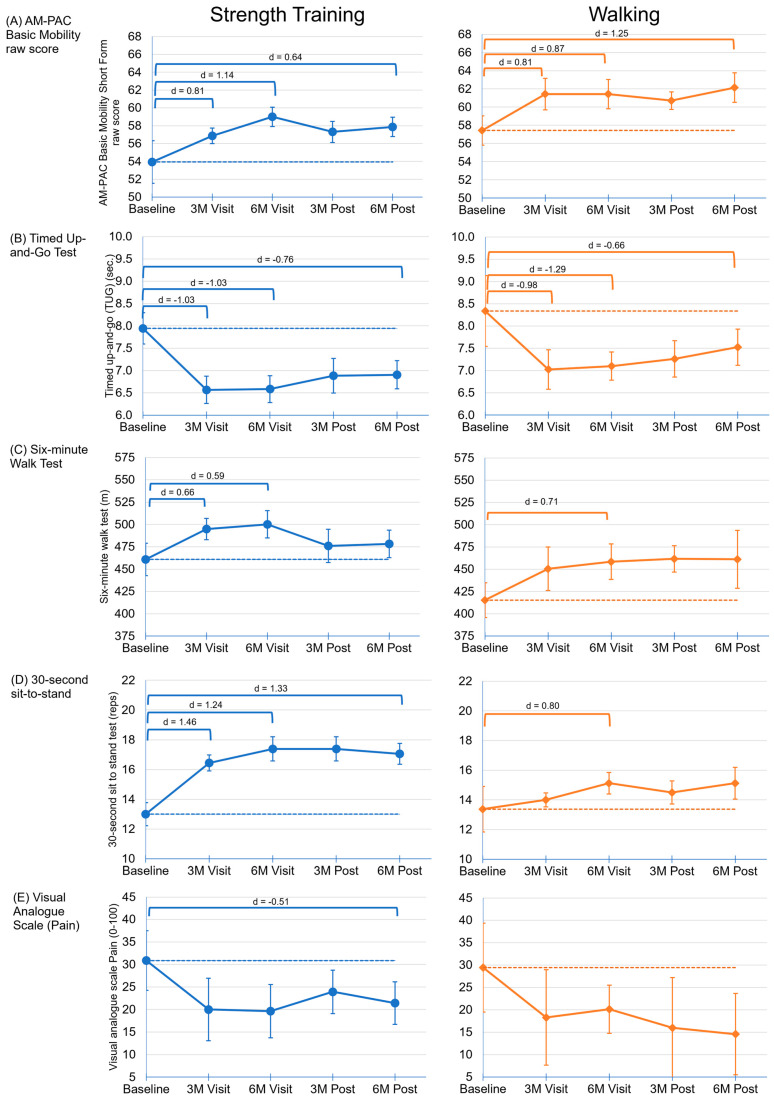
Sample mean values by timepoint for (**A**) AM-PAC basic mobility short form raw score; (**B**) timed up-and-go (TUG) test (seconds); (**C**) six-minute walk test (meters); (**D**) 30-second sit-to-stand test (reps); (**E**) visual analog scale (VAS) for pain. For (**B**,**E**), lower scores indicate better physical function; for (**A**,**C**,**D**), higher scores are better. The error bars are the standard errors from the generalized estimating equations (GEEs). The dashed lines represent the baseline average. The brackets represent statistically significant differences between timepoints from the GEE models. The effect size (d) is shown above the brackets. All measures only include the participants who completed all measures at all five timepoints (*n* = 18 STG; *n* = 8 WG) except AM-PAC: (*n* = 16 STG; *n* = 7 WG) and VAS: (*n* = 14 STG; *n* = 7 WG).

**Table 1 cancers-16-01774-t001:** Descriptive and disease characteristics of 42 participants who completed baseline visits.

	All Participants (*n* = 42)	Strength (*n* = 24)	Walking (*n* = 18)	*p*-Value ^1^
	Mean (SD)			
Age at consent (years)	63.2 (7.7)	63.9 (5.9)	62.2 (9.6)	0.52
Body mass index, baseline (BMI) (kg/m^2^)	30.3 (6.3)	30.9 (6.5)	29.5 (6.1)	0.49
	*n* (%) †			
Sex				
Female	26 (61.9)	14 (58.3)	12 (66.7)	0.58
Male	16 (38.1)	10 (41.7)	6 (33.3)	
Race and ethnicity				
Non-Hispanic White	36 (85.7)	21 (87.5)	15 (83.3)	0.66
Non-Hispanic Black or African-American	3 (7.1)	1 (4.2)	2 (11.1)	
Other or unknown race/ethnicity	3 (7.1)	2 (8.3)	1 (5.6)	
ECOG, baseline				
0 Fully active	26 (61.9)	17 (70.8)	9 (50.0)	0.26
1 Restricted in physically strenuous activity	15 (35.7)	7 (29.2)	8 (44.4)	
Disease response, baseline				
Complete response (CR)	26 (61.9)	16 (66.7)	10 (55.6)	0.44
<Complete response	15 (35.7)	8 (33.3)	7 (38.9)	
Disease status				
Active disease	4 (9.5)	1 (4.2)	3 (16.7)	0.17
Non-active disease	38 (90.5)	23 (95.8)	15 (83.3)	
Treatment at baseline ^2^				
None	6 (14.3)	4 (16.7)	2 (11.1)	0.11
Immunomodulatory drugs (IMiDs)	19 (45.2)	14 (58.3)	5 (27.8)	
IMiD + monoclonal antibody (MoAb)	7 (16.7)	1 (4.2)	6 (33.3)	
IMiD + proteosome inhibitor (PI)	5 (11.9)	3 (12.5)	2 (11.1)	
MoAb	1 (2.4)	1 (4.2)	0 (0.0)	
PI	3 (7.1)	1 (4.2)	2 (11.1)	
PI + MoAb	1 (2.4)	0 (0.0)	1 (5.6)	

^1^ Tested using Student’s *t*-test for the differences in the means or the Chi-squared test of differences in proportion. ^2^ All regimens except for the maintenance treatment with IMiDs or PIs also contained a glucocorticoid. † Total categorical values may not sum to 100% due to missing data and rounding. SD = standard deviation; *n* = number; ECOG = Eastern Cooperative Oncology Group performance status; CR = complete response.

**Table 2 cancers-16-01774-t002:** Adverse events reported during the physical activity intervention.

	Total Adverse Events	Grade 3+ ^1^	Related ^2^
Category	*N* (%)	*N* (%)	*N* (%)
Musculoskeletal and connective tissue disorders	42 (38.5)	0 (0)	18 (16.5)
Infections and infestations	18 (16.5)	1 (0.9)	0 (0)
Gastrointestinal disorders	9 (8.3)	0 (0)	1 (0.9)
Nervous system disorders	9 (8.3)	0 (0)	4 (3.7)
Injury, poisoning, and procedural complications	8 (7.3)	0 (0)	0 (0)
Respiratory, thoracic, and mediastinal disorders	8 (7.3)	0 (0)	0 (0)
General disorders and administration site conditions	6 (5.5)	0 (0)	0 (0)
Surgical and medical procedures	3 (2.8)	1 (0.9)	0 (0)
Vascular disorders	3 (2.8)	0 (0)	1 (0.9)
Skin and subcutaneous tissue disorders	1 (0.9)	0 (0)	0 (0)
Psychiatric disorders	1 (0.9)	0 (0)	0 (0)
Renal and urinary disorders	1 (0.9)	0 (0)	0 (0)
**All Categories**	**109 (100.0)**	**2 (1.8)**	**24 (22.0)**

^1^ Grade 3+ = at least severe or medically significant. Refer to Appendix A for a full description of adverse event grading and relatedness. ^2^ An adverse event was considered “related” if it was possible (*n* = 18), probable (*n* = 4), or definite (*n* = 2) that it was related. It was considered “unrelated” if it was unrelated (*n* = 62) or unlikely (*n* = 23) to be related. CTCAE = Common Terminology Criteria for Adverse Events.

**Table 3 cancers-16-01774-t003:** Serious adverse events in detail.

Category	Description of Serious Adverse Event	Severity	Relatedness
Gastrointestinal disorders	Gastrointestinal hemorrhage	5 (Death)	1 (Unrelated)
Infections and infestations	Appendicitis	3 (Severe)	1 (Unrelated)
Infections and infestations	Pneumonia, hospitalization	3 (Severe)	2 (Unlikely)
Infections and infestations	COVID-19 pneumonia	3 (Severe)	1 (Unrelated)
Infections and infestations	COVID-19 pneumonia (shortness of breath), further hospital admission	3 (Severe)	1 (Unrelated)
Surgical and medical procedures	Prolonged hospitalization after CAR-T cell therapy	3 (Severe)	1 (Unrelated)

## Data Availability

The raw data supporting the conclusions of this article will be made available by the authors on request.

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
