# Peer review of "Effects on the Physical Functioning of Two Exercise Interventions in Patients with Multiple Myeloma: A Pilot Feasibility Study"

_cancers, 2024, doi:10.3390/cancers16091774_

Round 1

Reviewer 1 Report

Comments and Suggestions for Authors

The paper concerns Feasibility and Effects on Physical Functioning of Two Exercise 2 Interventions in Patients with Multiple Myeloma it contains 2 tables no major grammer/spelling errors detected. physical functioning examined as well. methods are also given.

the paper is well written and nicely formatted statistical methods are proper tables and 1 figure are included number of references is proper the data is available upon request. funding, author contribution and competing interest statement included as well.    no ethics statement included

Author Response

Reponses to Reviewer’s Comments

(Note: The following is a duplicate of the attachment "Responses to reviewer 1 4-23-24."  This was done because the attachment seemed to disappear.)

The following presents our detailed responses to comments provided by Reviewer 1, with line numbers referring to both the tracked and clean versions.  We appreciate the careful review and suggestion provided by this expert and have endeavored to respond to it appropriately.

Reviewer #1

Comments and Suggestions for Authors

The paper concerns Feasibility and Effects on Physical Functioning of Two Exercise 2 Interventions in Patients with Multiple Myeloma it contains 2 tables no major grammer/spelling errors detected. physical functioning examined as well. methods are also given.

1) the paper is well written and nicely formatted statistical methods are proper tables and 1 figure are included number of references is proper the data is available upon request. funding, author contribution and competing interest statement included as well.    no ethics statement included

Response: I have added the following: All procedures performed in this study were in accordance with the ethical standards of the Center’s IRB and with the Belmont Report. (lines 85-86 tracked; lines 79-80 clean)

Reviewer 2 Report

Comments and Suggestions for Authors

Feasibility and Effects on Physical Functioning of Two Exercise 2 Interventions in Patients with Multiple Myeloma

Hillengass et al

Title

Suggest rewording to

Effects on Physical Functioning of Two Exercise 2 Interventions in Patients with Multiple Myeloma: a pilot feasibility study

MATERIALS AND METHODS

1.       Section 2.1 Patients and Interventions

This is poorly laid out and hard to follow.

Please rename section to patients and setting. Include name of center, population (eg outpatient MM patients etc) and where and how patients were recruited.

ECOG score isn’t defined or mentioned

While this isn’t an RCT a figure similar to a consort diagram would be helpful, beginning with Numbers approached, consented and enrolled by intervention , those who dopped out at various stages etc.  It’s not clear from line 78 if that happened pre or post approaching the patient for recruitment.

Add another section called ‘intervention’ with a description of each intervention

Line 70 i this isn’t dependent on it being a non-randomized design, as that could have been built into the study design. Just say something like ‘participants could take part in the intervention via zoom or equivalent if they lived too far from the institution to take part in the in-person group sessions’

Question: were these ‘remote’ ppts more likely to drop out? Maybe add to the adherence section

2.       Adverse events

Add a section on adverse events coding and what the scores mean

3.       Feasibility

The primary endpoint of feasibility was defined as the proportion of enrolled patients 83 who were still on study at the end of six months. (line 83)  This definition is limited  - retention of ppts who complete assessments doesn’t imply that the intervention is feasible in terms of interest etc. Clearly your study is feasible, but it would be good to include other criteria. These could include willingness of ppts to take part (you have these data), time needed to collect data (Fitbit might be an example of what wasn’t feasible) Practicality of delivering the interventions. Perhaps put into a table? Safety is another important component of this study that could be part of the feasibility

Did you interview or administer a questionnaire to ppts assess acceptability?  

Bowen et al Am J Prev med 2009 has a nice introduction

Table 1. Add a column on ‘all patients’

RESULTS

Start results section with

Of a total of 87 patients who were asked to participate, 43 signed the consent and  were enrolled.  Reasons for …etc.  Patients were %female with an average BMI of Z. etc. there were no differences in patient characteristics between arms.

Mention somewhere the number of ppts by arm who enrolled in the remote activities

Question: did you find out why ppts preferred an intervention over the other?

1.       feasibility and adherence

line 129 - 100% of the patients who were still on study at the six-month visit (79% of the 129 enrolled) performed all required interventions, however the aforementioned delays 130 caused the intervention period to stretch out beyond the anticipated six-month period; 131 only 32% of patients completed at least 80% of the training sessions within six months of 132 beginning the intervention

This is confusing. Break into (1) numbers completed (eg numbers on trial at different timepoints) and (2) the level of adherence at each timepoint to exercise goals: some ppts may have completed the assessments but may not have performed any exercise – currently that’s hard to parse from the way it’s presented. Perhaps integrate the adherence into table 2 –

2.       adverse events  Table 3. Spell out AE in table. Why would nervous system disorders be related to the intervention?  Again, footnote unhelpful without a section in the methods describing coding.  And what’s Grade 3?

3.       Physical functioning

·         Redefine the abbreviations somewhere in section 3,3.

·         Impossible to see graphs in detail – too small

·         Include a table of the results starting with mean VAS, grip strength at baseline, Timepoint 1, 2, 3 etc

·         Drop the combined charts – this is meaningless in the context of the study.

·         Also, I’m not sure the t-test is appropriate – this is a longitudinal analysis, and a GEE model would be better. Please consult with a statistician

DISCUSSION

1.       The start is a bit abrupt – also remind readers of what the 2 interventions were, and that this is a feasibility study.

2.       Line 197.  It is now recommended that researchers move away from statistical significance as it is over interpreted, and that the focus be on effect size. From your data it’s clear there are differences in effects. I think you can rephrase this line to read ‘the effects demonstrate difference in XXX  between the WG and STG. While these did not reach statistical significance, this was a feasibility study and as such was not powered to detect differences between arms’ (or something to that effect)

3.       Line 203 For obvious reasons in exercise trials, it is not possible to blind participants to randomiza tion. Please remove. Blinding in RCTs testing behavioral interventions refers to investigator and statistician blinding, not the participants.

4.       Line 204 Additionally, defining an intervention for a control group is challenging in exercise 204 interventions with cancer patients, since even interventions like relaxation techniques or 205 general health education can influence individual behavior, and prescribed PT as stand- 206 ard of care itself can limit the significance of findings in the intervention group

Please remove. This is the whole point of controls. If you do a relaxation intervention as a control and then find no difference between it and strength training, then why do the strength training. That is equivalent to not testing a standard of care  against a test drug as it can limit the significance of findings in the intervention group

There are many reasons not to include a control group, but this isn’t one of them. For example you could conduct a non-inferiority test to show that a walking or homebased ST is at least as good as PT training – therefore making it more accessible (or whatever)

5.       There’s no need to explain why there wasn’t a waitlist – this is a pilot feasibility study, and that’s fine – in the larger study, you could discuss that however.

6.       Lines 223 onwards – please add some details on what type of interventions – eg were they strength training? Aerobic? Short or long term? They might differ significantly from this study. (actually, I see these issues are raised a bit further down) This is an interesting section, and I’d move lines 223-245 to just below line 194

7.       Just before the conclusions add a strengths and limitations section –

Eg the main limitation was the small sample size, and the fact that there wasn’t a control arm. Then you can add that you still saw an effect, and as this was a feasibility study controls not needed. (or something like that)

Author Response

Reponses to Reviewer’s Comments

(Note: The following is a duplicate of the attachment "Responses to reviewer 2 4-23-24."  This was done because the attachment seemed to disappear.)

The following presents our detailed responses to comments provided by Reviewer 2, with line numbers referring to both the tracked and clean versions.  We appreciate the careful review and suggestions provided by this expert and have endeavored to respond to each with thoroughness.

Reviewer #2

Title

Suggest rewording to

Comment #1: Effects on Physical Functioning of Two Exercise 2 Interventions in Patients with Multiple Myeloma: a pilot feasibility study

Response #1: We have reworded the title accordingly.  Thank you.

MATERIALS AND METHODS

  1. Section 2.1 Patients and Interventions

This is poorly laid out and hard to follow.

Comment #2: Please rename section to patients and setting. Include name of center, population (eg outpatient MM patients etc) and where and how patients were recruited.

Response #2: We have renamed the section and added these details to the first paragraph. (lines 77-83 tracked; lines 72-76 clean)

Comment #3: ECOG score isn’t defined or mentioned

Response #3: A line was added to say that there were no differences in patient characteristics by treatment arm, including in ECOG performance status, which we’ve spelled out here (lines 194, 201-204 tracked; lines 162, 169-172 clean).

Comment #4: While this isn’t an RCT a figure similar to a consort diagram would be helpful, beginning with Numbers approached, consented and enrolled by intervention , those who dopped out at various stages etc.  It’s not clear from line 78 if that happened pre or post approaching the patient for recruitment.

Response #4:  Thank you for that suggestion.  We have added a flowchart (Figure 1) showing participation at multiple timepoints, first referred to at line 114 tracked, line 93 clean.  Regarding line 78, the version I submitted has different line numbers.  I believe this refers to the line about imaging results being reviewed by the tumor board.  I have added a sentence explaining that pre-screening was done by the treating physician and that tumor board review was done after remaining patients who were approached were screened and considered eligible. (lines 105-108 tracked; lines 84-87 clean)

Comment #5: Add another section called ‘intervention’ with a description of each intervention.

Response #5: We have created an Intervention section and added some detail about the strength training intervention (lines 121-128 tracked, lines 100-107 clean).

Comment #6: Line 70 i this isn’t dependent on it being a non-randomized design, as that could have been built into the study design. Just say something like ‘participants could take part in the intervention via zoom or equivalent if they lived too far from the institution to take part in the in-person group sessions’

Response #6: Sorry for the confusion, but there was no Zoom option for the STG. The STG was in person at our hospital, and the walking group was at home, with prompts delivered remotely to their Fitbits to prompt them to increase their activity.  We have clarified this in the intervention description (lines 128-131 tracked; lines 107-110 clean).

Comment #7: Question: were these ‘remote’ ppts more likely to drop out? Maybe add to the adherence section

Response #7: The WG was not more likely to drop out than the STG.  We added a p value to the adherence section (line 237 tracked; line 201 clean).

  1. Adverse events

Comment #8: Add a section on adverse events coding and what the scores mean

Response #8: We have added detail to this section and included Supplementary Tables S1-S2 listing the categories and describing the grades and relationships of the AEs to the intervention (lines 133-141 tracked, lines 112-120 clean).

  1. Feasibility

Comment #9: The primary endpoint of feasibility was defined as the proportion of enrolled patients 83 who were still on study at the end of six months. (line 83)  This definition is limited  - retention of ppts who complete assessments doesn’t imply that the intervention is feasible in terms of interest etc. Clearly your study is feasible, but it would be good to include other criteria. These could include willingness of ppts to take part (you have these data), time needed to collect data (Fitbit might be an example of what wasn’t feasible) Practicality of delivering the interventions. Perhaps put into a table? Safety is another important component of this study that could be part of the feasibility

Did you interview or administer a questionnaire to ppts assess acceptability? 

Bowen et al Am J Prev med 2009 has a nice introduction

Response #9: The proportion on the study at six months was our definition of feasibility, even though this measure also speaks to adherence.  We have provided the number of patients who signed the consent, i.e., the willingness to take part, in the Feasibility and Adherence section.  We have mentioned Fitbit syncing issues in the Feasibility and Adherence section, and we have added that this impacted our reporting of their progress. The major impediment to the success of the in-person arm was the Covid-19 concerns caused interruptions and some patients even chose not to continue after interruptions.   We have added some language clarifying all of this to the Feasibility and Adherence section.  We have also added a note about our ability to obtain the PT/physical function assessments for all the timepoints of interest.  We did not administer a questionnaire to participants to assess acceptability (lines 223-237, 251-253 tracked; lines 190-201, 215-217 clean).

Comment #10: Table 1. Add a column on ‘all patients’

Response #10: We have added this column to Table 1.

RESULTS –

Comment #11:

Start results section with

Of a total of 87 patients who were asked to participate, 43 signed the consent and  were enrolled.  Reasons for …etc.  Patients were %female with an average BMI of Z. etc. there were no differences in patient characteristics between arms.

Mention somewhere the number of ppts by arm who enrolled in the remote activities

Response #11:  We placed the Table 1 details at the start of the Results section (lines 190-201 tracked; lines 158-168 clean).  We have added Figure 1 to show numbers of patients at each timepoint.

Comment #12:

Question: did you find out why ppts preferred an intervention over the other?

Response #12: We did not collect data from each participant on why they chose the strength training group or walking group.  Our sense is that many of them preferred the walking group because it enabled them to be remote, which meant that it was more convenient and did not create any Covid concerns.

  1. feasibility and adherence

Comment #13:

line 129 - 100% of the patients who were still on study at the six-month visit (79% of the 129 enrolled) performed all required interventions, however the aforementioned delays 130 caused the intervention period to stretch out beyond the anticipated six-month period; 131 only 32% of patients completed at least 80% of the training sessions within six months of 132 beginning the intervention

 This is confusing. Break into (1) numbers completed (eg numbers on trial at different timepoints) and (2) the level of adherence at each timepoint to exercise goals: some ppts may have completed the assessments but may not have performed any exercise – currently that’s hard to parse from the way it’s presented. Perhaps integrate the adherence into table 2 –

Response #13:

Sorry for the confusion.  Our secondary endpoint of adherence within the STG was measured by “the proportion of patients remaining on study at six months who completed 80% or more of the strength training intervention activities.”  The point of the statement that only 32% completed at least 80% of the training sessions within six months was to quantify the impact of delays (due mostly to Covid concerns and medical issues that arose).  But they eventually completed 100% of the strength training sessions.  We have added the range and median months it took for the STG to complete the six-month intervention in order to highlight the fact that the intervention period stretched out longer than six months (line 243 tracked, line 206 clean). We hope this clarifies it.

Additionally, Table 2 was not extremely useful in its original form, so it was deleted.  It did not show participation by timepoint.  It only showed the count of patients who had PT/functional assessments done at each point.  It was possible for a patient to be on the study but to have missed the PT assessment at a given timepoint.  We have created a flowchart  (Figure 1) which shows, for each arm and timepoint, how many participants there were (n=) and how many physical function assessments (a=) there were from patients participating in the study at that timepoint.

Comment #14:

  1. adverse events Table 3. Spell out AE in table. Why would nervous system disorders be related to the intervention?  Again, footnote unhelpful without a section in the methods describing coding.  And what’s Grade 3?

Response #14.  We have spelled out AE in the table and added a footnote explaining grading to Table 3.  Regarding nervous system disorders, the CTCAE term “Dizziness” falls under this category in CTCAE version 5. We have added an explanation of this to the results.  We have also explained the AE reporting hierarchy in the Methods under the Safety section. (lines 133-143, 260-261 tracked; lines 112-123, 224-225 clean).

  1. Physical functioning

Comment #15: Redefine the abbreviations somewhere in section 3,3.

Response #15: We have redefined the functional measurements in this section (lines 282-299 tracked; lines 241-251 clean).

Comment #16: Impossible to see graphs in detail – too small

Response #16: I have remade these graphs and increased font sizes, improved colors, and provided a more informative X axis.  I have also removed the combined charts. We hope this improves the visual experience.

Comment #17:Include a table of the results starting with mean VAS, grip strength at baseline, Timepoint 1, 2, 3 etc

Response #17: Thank you for this comment, but we have decided to stay with the visual representation, in order to avoid redundant information and to make it easier to see how most measures were improved, and most improvements were sustained.

Comment #18: Drop the combined charts – this is meaningless in the context of the study.

Response #18: We have done this in the revised figure, now called Figure 2.  Thank you.

Comment #19: Also, I’m not sure the t-test is appropriate – this is a longitudinal analysis, and a GEE model would be better. Please consult with a statistician.

Response #19: At your advice and the advice of a statistician (K. Attwood), we have changed these to GEE models and added the statistician to our study team / author list.  After that change, there were additional significant timepoints, as described in the revised manuscript. Thank you.  (lines 180-181 tracked, lines 149-150 clean).

DISCUSSION

Comment #20:

  1. The start is a bit abrupt – also remind readers of what the 2 interventions were, and that this is a feasibility study.

Response #20:

Thank you for pointing this out.  We have reworked the first few sentences of the Discussion to include this information and be less abrupt (lines 318-327 tracked; lines 267-271 clean).

Comment #21:

  1. Line 197. It is now recommended that researchers move away from statistical significance as it is over interpreted, and that the focus be on effect size. From your data it’s clear there are differences in effects. I think you can rephrase this line to read ‘the effects demonstrate difference in XXX  between the WG and STG. While these did not reach statistical significance, this was a feasibility study and as such was not powered to detect differences between arms’ (or something to that effect)

Response #21:

We have added some verbiage on the magnitude of effects being similar in most measures, and emphasized that the pilot nature of the study led to small sample sizes. We have also added in effect sizes to the physical function results section. (lines 182-184, 339-342 tracked; lines 151-152, 285-289 clean).

Comment #22

  1. Line 203 For obvious reasons in exercise trials, it is not possible to blind participants to randomiza tion. Please remove. Blinding in RCTs testing behavioral interventions refers to investigator and statistician blinding, not the participants.

Response #22

We agree and have removed this sentence (lines 352-353 tracked; after line 294 clean).  Thank you.

Comment #23

  1. Line 204 Additionally, defining an intervention for a control group is challenging in exercise 204 interventions with cancer patients, since even interventions like relaxation techniques or 205 general health education can influence individual behavior, and prescribed PT as stand- 206 ard of care itself can limit the significance of findings in the intervention group

Please remove. This is the whole point of controls. If you do a relaxation intervention as a control and then find no difference between it and strength training, then why do the strength training. That is equivalent to not testing a standard of care  against a test drug as it can limit the significance of findings in the intervention group

There are many reasons not to include a control group, but this isn’t one of them. For example you could conduct a non-inferiority test to show that a walking or homebased ST is at least as good as PT training – therefore making it more accessible (or whatever)

Response #23

We agree and have removed this sentence (lines 353-356 tracked; after line 294 clean).  Thank you.

Comment #24

  1. There’s no need to explain why there wasn’t a waitlist – this is a pilot feasibility study, and that’s fine – in the larger study, you could discuss that however.

Response #24

We have removed the sentences about the waitlist control group (lines 356-362 tracked; after line 294 clean).

Comment #25

  1. Lines 223 onwards – please add some details on what type of interventions – eg were they strength training? Aerobic? Short or long term? They might differ significantly from this study. (actually, I see these issues are raised a bit further down) This is an interesting section, and I’d move lines 223-245 to just below line 194

Response #25

Thank you for the suggestion.  With all the edited text and reorganization, we have decided to leave this section where it is.  

Comment #26

  1. Just before the conclusions add a strengths and limitations section –

Eg the main limitation was the small sample size, and the fact that there wasn’t a control arm. Then you can add that you still saw an effect, and as this was a feasibility study controls not needed. (or something like that)

Response #26

Thank you for this suggestion.  I have added a paragraph with strengths and limitations right before the conclusion (lines 401-416 tracked; lines 332-347 clean).

Reviewer 3 Report

Comments and Suggestions for Authors

Patients with multiple myeloma received interventions, which were personal resistance training, or encouragement for 150–300 minutes active walk. The interventions were assessed for feasibility to adhere to the instruction for six months, and 74% of participants completed 6-month trainings. Additionally, activity of daily living, motor functions, and pain were assessed in comparison with those at the baseline. This is a pilot study maybe for a future large study, but the background was not elaborated, and the hypothesis is vague. Specified comments are here.

1.     Lines 46–50. “Certain MM therapies, especially ASCT and CAR-T, are associated with some level of toxicity, and patient frailty is considered a risk factor for complications, sometimes leading to exclusion of patients from these therapies. Frailty is, however, a somewhat influenceable factor and can be improved by interventions like PT and physical exercise.” There are no references. It is unknown why “frailty” is an exclusion criterion for therapies, such as ASCT and CAR-T. The disease status, and ECOG of most participants were non-active, and normal with no limitation for activity. This may indicate that participants had been diagnosed before, received treatment for multiple myeloma, and have achieved a remission to some extent. This stage may not be decision of a therapy. What is influenced by frailty? Or, is improvement of frailty intervention outcome? The background should be described in detail, citing appropriate literatures.

2.     Line 52. This is a pilot study, so the authors should clarify what is needed for a main study in this pilot study. The results described in Lines 154–171 exhibited p values. Rather than that, amounts of changes with standard deviation (not standard errors) are needed to calculate a sample size for a main study. Figure 1 is too small to read text within it. The readers cannot read effect sizes.

3.     Patient characteristics are shown in Table 1 (Line72). Based on the background of this study (that is, motor functions are important for decision of therapy options), therapy type, phase of treatment, and bone involvement are needed. Bone and spine stability was assessed before the interventions. Does no one have bone involvement? A breakdown list should be exhibited, “The patient was either cleared for participation, cleared with limitations, or declined for participation.”

4.     Lines 92–102. Physical therapy assessment. Based on the background of this study (that is, to improve frailty), define “frailty”, and explain how assessment tools are related to frailty, or differentiate them between the related and non-related ones.

5.     Additionally, clarify scoring and measuring of assessment with references for each one. For AM-PAC “lower scores equate to lower levels of function,” in Jette et al. 2014. However, AM-PAC score decreased in Figure 1 despite the scores improved according to the text. In addition, clarify what domain was assessed in AM-PAC.

6.     Lines 106–107. “Measures of physical function by timepoint were compared using paired T tests for differences in mean.” In this case, test is multi-comparison. Tukey’s test is better than paired t test.

Minor points

7.     Lines 183–184. “The current pilot study shows improvement of parameters of physical functioning after either of two six-month exercise interventions.” The main aim of this study was to investigate feasibility of interventions. This should be described first.

8.     Lines 204–207. “Additionally, defining an intervention for a control group is challenging in exercise interventions with cancer patients, since even interventions like relaxation techniques or general health education can influence individual behavior, and prescribed PT as standard of care itself can limit the significance of findings in the intervention group.“ This is a reason why a control trial should be implemented. This is not an explanation that it is difficult to implement controlled trial. The authors can compare rigorous wiht mild interventions. Delete this sentence.

9.     Lines 144, and 221. What is grade?

10.   Lines 135–139. Fonts were changed.

11.   Line 229. Parenthesis.

12.   This study has been reported previously in the International Myeloma Society 20th Annual Meeting and Exposition. It has a DOI number, 10.1016/S2152-2650(23)01605-1. This should be mentioned in the backmatter.

Author Response

Responses to Reviewer’s Comments

(Note: The following is a duplicate of the attachment "Responses to reviewer 3  4-23-24." This was done because the attachment seemed to disappear.)

The following presents our detailed responses to comments provided by Reviewer 3, with line numbers referring to both the tracked and clean versions.  We appreciate the careful review and suggestions provided by this expert and have endeavored to respond to each with thoroughness.

Reviewer #3

Comments and Suggestions for Authors

Patients with multiple myeloma received interventions, which were personal resistance training, or encouragement for 150–300 minutes active walk. The interventions were assessed for feasibility to adhere to the instruction for six months, and 74% of participants completed 6-month trainings. Additionally, activity of daily living, motor functions, and pain were assessed in comparison with those at the baseline. This is a pilot study maybe for a future large study, but the background was not elaborated, and the hypothesis is vague. Specified comments are here.

  1. Lines 46–50. “Certain MM therapies, especially ASCT and CAR-T, are associated with some level of toxicity, and patient frailty is considered a risk factor for complications, sometimes leading to exclusion of patients from these therapies. Frailty is, however, a somewhat influenceable factor and can be improved by interventions like PT and physical exercise.” There are no references. It is unknown why “frailty” is an exclusion criterion for therapies, such as ASCT and CAR-T. The disease status, and ECOG of most participants were non-active, and normal with no limitation for activity. This may indicate that participants had been diagnosed before, received treatment for multiple myeloma, and have achieved a remission to some extent. This stage may not be decision of a therapy. What is influenced by frailty? Or, is improvement of frailty intervention outcome? The background should be described in detail, citing appropriate literatures.

Response: Thank you for this feedback.  We have added explanations that frailty can influence treatment dose, duration, and outcomes and provided a reference.  We have also added the fact that frailty is common in MM, especially in relapsed/refractory disease, and can limit treatment options and result in worse outcomes, so interventions that improve frailty can set the stage for better outcomes if a patient relapses and needs additional treatment. (lines 64-69 tracked; lines 59-64 clean)

  1. Line 52. This is a pilot study, so the authors should clarify what is needed for a main study in this pilot study. The results described in Lines 154–171 exhibited p values. Rather than that, amounts of changes with standard deviation (not standard errors) are needed to calculate a sample size for a main study. Figure 1 is too small to read text within it. The readers cannot read effect sizes.

Response: We have changed these from p-values to effect sizes (mean and st.dev of differences by timepoint) and improved the visualization of Figure 1 (now figure 2). We also hope that the journal will use a higher res version of the figure.  Please note: the other reviewer suggested changing from paired T-tests to GEE, which we have decided to do because we’re modeling longitudinal data.  After that change, there were additional significant timepoints, as described in the revised manuscript (lines 182-184, 274-275 tracked; lines 151-152, 236-237 clean).

  1. Patient characteristics are shown in Table 1 (Line 72). Based on the background of this study (that is, motor functions are important for decision of therapy options), therapy type, phase of treatment, and bone involvement are needed. Bone and spine stability was assessed before the interventions. Does no one have bone involvement? A breakdown list should be exhibited, “The patient was either cleared for participation, cleared with limitations, or declined for participation.”

Response: Thanks for this important suggestion. We have added current treatments being received by the participants to Table 1 and the results section.  We have also added tumor board recommendations about limitations to Table 1 and the results section (lines 195-201 tracked; lines 163-169 clean).

  1. Lines 92–102. Physical therapy assessment. Based on the background of this study (that is, to improve frailty), define “frailty”, and explain how assessment tools are related to frailty, or differentiate them between the related and non-related ones.

Response: Although an important state, frailty was not used as a defined state or a specific score in our study. Although there is an IMWG frailty score, we do not have all the information for that. Additionally, although a lot of myeloma patients are frail, we didn’t specifically target frail patients for inclusion in the pilot study. We have added something to the discussion about the IMWG frailty score, which we plan to use (or something similar to it) in future studies (lines 335-337 tracked, lines 282-284 clean).

  1. Additionally, clarify scoring and measuring of assessment with references for each one. For AM-PAC “lower scores equate to lower levels of function,” in Jette et al. 2014. However, AM-PAC score decreased in Figure 1 despite the scores improved according to the text. In addition, clarify what domain was assessed in AM-PAC.

Response: We have references for all the PT measures. With regard to the AM-PAC basic mobility score, we use a converted score in our clinic which translates the AM-PAC raw score (which, as you pointed out, means worse mobility when the number is lower) to a converted score scaled from 0% to 100% impairment based on a Centers for Medicare and Medicaid (CMS) conversion tables, so higher numbers are worse.  However, to be consistent with what is more widely recognized in the literature, we have changed this to the raw score, meaning that higher scores are better. This has been reflected in Figure 2 and the Results section (lines 275-279 tracked; lines 237-241 clean). 

  1. Lines 106–107. “Measures of physical function by timepoint were compared using paired T tests for differences in mean.” In this case, test is multi-comparison. Tukey’s test is better than paired t test.

Response: Thank you for pointing this out.  Another reviewer also objected to our use of the paired t test and suggested that we use generalized estimating equation (GEE) models.  Because this is a longitudinal study, we have changed our models to GEE and updated the results accordingly.  Going from the paired t tests to the GEE models resulted in a few additional significant findings, reflected in the Results section.

Minor points

  1. Lines 183–184. “The current pilot study shows improvement of parameters of physical functioning after either of two six-month exercise interventions.” The main aim of this study was to investigate feasibility of interventions. This should be described first.

Response: We have reworked the early part of the Discussion to focus first on feasibility and adherence and then on the physical function measures (lines 318-322 tracked; lines 267-271 clean).

  1. Lines 204–207. “Additionally, defining an intervention for a control group is challenging in exercise interventions with cancer patients, since even interventions like relaxation techniques or general health education can influence individual behavior, and prescribed PT as standard of care itself can limit the significance of findings in the intervention group.“ This is a reason why a control trial should be implemented. This is not an explanation that it is difficult to implement controlled trial. The authors can compare rigorous wiht mild interventions. Delete this sentence.

Response: We have deleted this section (lines 351-362 tracked; after line 294 clean).

  1. Lines 144, and 221. What is grade?

Response: We have added better descriptions of the adverse event criteria (grade/relationship) to the methods/intervention section (lines 133-144 tracked; lines 112-123 clean).

  1. Lines 135–139. Fonts were changed.

Response: I believe we have fixed this now.

  1. Line 229. Parenthesis.

Response: I am not sure what lines this refers to, as the version I submitted has different line numbers, but I hope I caught it in other editing.

  1. This study has been reported previously in the International Myeloma Society 20th Annual Meeting and Exposition. It has a DOI number, 10.1016/S2152-2650(23)01605-1. This should be mentioned in the backmatter.

Response: Thank you for catching that.  We have added this to Funding and Acknowledgements section (lines 447-449 tracked; lines 378-380 clean).

Submission Date

27 February 2024

Date of this review

12 Mar 2024 09:54:05

Round 2

Reviewer 2 Report

Comments and Suggestions for Authors

Thank you for the thorough response to my comments. I think the paper is much improved

A few small edits -

1. section 3.1 - White - this should be categorized as Non-Hispanic White (if that is the case) as per US census categorization. Table 1 is missing ethnicity categories - please add. 

2. Under methods - statistics

Measures of physical function by timepoint were compared using generalized estimating equations.

It is more correct to say 'we compared changes in phys functioning  outcomes over time between groups, using the generalized estimating equations (GEE) modification of linear regression to account for the correlation within individuals over time.'

3. Under results

line 169 There were no differences by treatment arm in these characteristics, with the exception of tumor board clearance, as a subset of the STG participants were required to receive neurosurgery or orthopedic clearance to participate.

I think drop this statistical test - it's not a very meaningful comparison as you describe. Perhaps take it out of the table, and just say that more STG patients required clearance, and move up under participant selection where you say that no patients were excluded after tumor board review

4. Results

line 202 Adherence to the requirements of the study was generally very good and was
stronger in the STG than in the WG.

You can remove this sentence - qualifiers  like these should be in discussion not results sections. Just report the adherence for the 2 groups .

The following sentence is long- split into two

5. Figure 2

Much improved - thanks!

Can you add the effect size  over each bracket please?

6. Discussion

a. in first sentence ' immune profile,' is mentioned. specify that this is the subject of a different paper

b. Although there was a tendency to more significant effects  - drop this. Compare only effect sizes. 'Tendancies to more significant effects' is sort of meaningless as this study wasn't powered to detect these

c. line 291  Furthermore, the more hands-off intervention in the WG led to a lower adherence, at least when objectively assessed based on active 292
minutes acquired from their wearable devices.

can you add some refs  please to say that unsupervised behavioural interventions usually have lower adherence

d. line 295 The exercise interventions in this study were performed...

should this be 'were designed'?

e.  recent review of feasibility and safety of exercise interventions

in MM patients? cancer patients in general? others?

f. line 306 was this study strength training or walking or something else?

can you Specifiy what the intervention was (one or two words is fine), especially as in line 312 you specify the intervention for a different study. And Coleman's findings (ref 34) aren't described here

Author Response

Reponses to Reviewer’s Comments, Round 2

The following presents our detailed responses to Round 2 comments provided by Reviewer 2.  Thank you for your diligent review!

Reviewer #2

Thank you for the thorough response to my comments. I think the paper is much improved

A few small edits -

  1. section 3.1 - White - this should be categorized as Non-Hispanic White (if that is the case) as per US census categorization. Table 1 is missing ethnicity categories - please add.

Response: I have reframed this to be non-Hispanic White, non-Hispanic Black, and other/unknown race/ethnicity.

  1. Under methods - statistics

Measures of physical function by timepoint were compared using generalized estimating equations.

It is more correct to say 'we compared changes in phys functioning  outcomes over time between groups, using the generalized estimating equations (GEE) modification of linear regression to account for the correlation within individuals over time.'

Response: We have made this edit.  Thank you for your guidance on this.

  1. Under results

line 169 There were no differences by treatment arm in these characteristics, with the exception of tumor board clearance, as a subset of the STG participants were required to receive neurosurgery or orthopedic clearance to participate.

I think drop this statistical test - it's not a very meaningful comparison as you describe. Perhaps take it out of the table, and just say that more STG patients required clearance, and move up under participant selection where you say that no patients were excluded after tumor board review

Response: We have removed this from the table and results section and moved the details up to the Patients and Setting section.

  1. Results

line 202 Adherence to the requirements of the study was generally very good and was

stronger in the STG than in the WG.

You can remove this sentence - qualifiers  like these should be in discussion not results sections. Just report the adherence for the 2 groups .

Response: We have changed this sentence accordingly.

The following sentence is long- split into two

Response: We have broken this sentence into two.

  1. Figure 2

Much improved - thanks!

Can you add the effect size  over each bracket please?

Response: We have added in the effect sizes for the significant differences and changed the legend accordingly.

  1. Discussion

  1. in first sentence ' immune profile,' is mentioned. specify that this is the subject of a different paper

Response: We have added a reference to this paper, which was just accepted.

  1. Although there was a tendency to more significant effects - drop this. Compare only effect sizes. 'Tendancies to more significant effects' is sort of meaningless as this study wasn't powered to detect these

Response: We have changed this sentence to: The two intervention groups showed improvements in all measures with a similar magnitude of effect, with the exception of the 30SST, which had markedly more pronounced and sustained improvement in the STG.

  1. line 291 Furthermore, the more hands-off intervention in the WG led to a lower adherence, at least when objectively assessed based on active 292

minutes acquired from their wearable devices.

can you add some refs  please to say that unsupervised behavioural interventions usually have lower adherence

Response: We have added a reference and some verbiage on a meta-analysis comparing adherence in unsupervised and supervised.  There was non-significantly higher adherence in supervised studies, but the authors noted that the unsupervised trainings’ adherence might have been inflated due to self-report and some aspect of supervision, issues our WG would not have been subject to.

  1. line 295 The exercise interventions in this study were performed...

should this be 'were designed'?

Response: Yes!  Thank you!

  1. recent review of feasibility and safety of exercise interventions

in MM patients? cancer patients in general? others?

Response: This was in patients with myeloma.  We have added the reference.  Thank you.

Nicol JL, Chong JE, McQuilten ZK, Mollee P, Hill MM, Skinner TL. Safety, Feasibility, and Efficacy of Exercise Interventions for People With Multiple Myeloma: A Systematic Review. Clin Lymphoma Myeloma Leuk. 2023 Feb;23(2):86-96. doi: 10.1016/j.clml.2022.10.003. Epub 2022 Oct 22. PMID: 36450625.

  1. line 306 was this study strength training or walking or something else?

Response: We have added these details.

can you Specifiy what the intervention was (one or two words is fine), especially as in line 312 you specify the intervention for a different study. And Coleman's findings (ref 34) aren't described here

Response: We have added more details about the other interventions.

Reviewer 3 Report

Comments and Suggestions for Authors

Thank you for your deligent revision and polite responses. The revision improves the manusricpt complihensive. I am looking forward to the main study report. Good luck. 

Line 150. Use P <0.05 instead of P ≤ 0.05.  

In statistical analysis and Figure 2c, I think that only comparison with the baseline is enough. 

Author Response

Reponses to Reviewer’s Comments, Round 2

The following presents our detailed responses to comments provided by Reviewer 3 in the second round.

Reviewer #3

Thank you for your diligent revision and polite responses. The revision improves the manuscript complihensive. I am looking forward to the main study report. Good luck.

Line 150. Use P <0.05 instead of P ≤ 0.05. 

In statistical analysis and Figure 2c, I think that only comparison with the baseline is enough.

Response: Thank you for all your advice!  We have changed this to a < sign.  We have also removed the bracket showing the significant drop in the 6MWT from 6M to 3P and adjusted the results accordingly to say that the improvement was not sustained.